# The Feasibility of Hydroxypropyl Methylcellulose as an Admixture for Porous Vegetarian Concrete Using Coarse Recycled Aggregates

**Kun Zhang [1], Wei Yin [1,\*], Xi Chen [2], Hui Li [3], Mingxing Cao [1] and Shengxue Zhu [1]**

[1] Department of Transportation Engineering, Huaiyin Institute of Technology, Huaian 223003, China; zhangkun89728@gmail.com (K.Z.); cmx786584043@163.com (M.C.); zsx10316@hyit.edu.cn (S.Z.)

[2] Department of Architecture and Civil Engineering, Huaiyin Institute of Technology, Huaian 223003, China; cricketchen@yeah.net

[3] Department of Transportation Engineering, Nanjing Tech University, Nanjing 211816, China; 202061228006@njtech.edu.cn

\* Correspondence: yinweihyit@163.com

**Abstract:** In this paper, hydroxypropyl methylcellulose (HPMC) is used as a new additive for porous vegetarian concrete (PVC) to improve its void structure and strength. The effect of the HPMC on the fluidity of the mortar was first investigated by a viscosity test. Then the cement hydration process was determined for analyzing the effect of the HPMC on the strength and durability of the hardened PVC. Subsequently, experiments to investigate the mass transport and compressive strength characteristics, as well as the vegetarian properties, of the concrete were carried out. The results show that the bonding forces between the recycled aggregates and packing layer are elevated by viscosity improvement. The viscocity and flowability are significantly related to the dosage of HPMC from 0.0‰ to 0.3‰. The harden time is also delayed while the content of HPMC increases. The segregation phenomenon caused by the recycled aggregate powder in porous concrete could also be relieved by adding HPMC. The durability of PVC in the wetting–drying cyclic test is significantly improved by incorporating HPMC. The results of the vegetarian test also prove that, with HPMC mixing, sufficient space would be created in porous concrete, which is more suitable for plant growth due to a large number of existing pore channels.

**Keywords:** porous vegetarian concrete; hydroxypropyl methylcellulose; viscosity; void structure; plant growth

## 1. Introduction

With urbanization in China increasing, a high amount of construction and demolition waste is produced and consumes large volumes of natural resources [1,2]. In this context, recycled aggregates (RA) have been recognized as preserving natural resources and reducing space for waste storage by replacing the natural aggregates with fresh concrete [3–5]. Recycling concrete is a relatively simple process, which involves demolition collection, crushing, and isolating [6,7]. Compared with natural aggregates, the production process causes a series of issues in the use of RA. The high powder content is one of the significant problems. Amounts of insoluble powder adhering to the surface of the particles were involved in fresh concrete when the mixing process was completed, which could cause cracking and segregating problems after the concrete was hardened [8–10]. Bendimerad et al. [11] assessed the parameters used to compare the cracking sensitivity of different concretes with recycled aggregates through a stress/strength experimental method. A high rate of substitution of recycled gravel or sand affected the early age properties of the recycled concrete and the cracking sensitivity, especially when natural sand was replaced by recycled concrete sand. Xiao et al. [12] revealed that the shear transfer mechanism and process across cracks in

RAC (recycled coarse aggregate) are largely the same as those in natural aggregate concrete (NAC). Both the lateral constraint and the concrete compressive strength positively affect the shear transfer strength of the RAC.

The above cracking phenomenon would be more critical in low mortar/aggregate ratio situations, such as porous concrete [13,14]. As one kind of many porous media, the strength is significantly affected by the porosity of its internal structure. The presence of pores can adversely affect the material?s mechanical properties, such as failure strength, elasticity and creep strains [15]. Porous concrete was first invented by a Japanese engineer in the 1980s as an environmentally friendly material [16]. Since then, it has been widely used in various applications in Japan, USA and Europe because of its multiple environmental benefits: controlling storm water runoff, restoring groundwater supplies, and reducing water and soil. Park et al. [17] experimented with two kinds of RAC and discovered that a porous concrete with a smaller size of aggregates and a higher void content have superior ability of reducing the total phosphorus (T-P, mg/L) and total nitrogen (T-N, mg/L) in the test water. Xu et al. [18] demonstrated that the total porosity of porous concrete shows a linear relation to the effective porosity, and a power function relation exists between permeability and effective porosity.

Due to its superior drainage and structural characteristics, porous concrete has been documented as an efficient material widely used in the rainy south of China for slope protection and landscaping building [19,20]. Porous concrete contains little or no fine aggregates, using an adequate amount of cement paste to coat and bind the coarse aggregates to create a system of high porosity and interconnected voids to drain off the water quickly. These superior properties are very suitable for plant growth [16,21], which was named porous vegetarian concrete (PVC). Hwang-Hee Kim and Chan-Gi Park [22] investigated that with latex used in PVC materials, the compressive strength, void ratio and freeze-thaw resistance all improved, satisfying the target performance metrics; additionally, vegetation growth tests showed that plant growth was more active. Quan and Hong Zhu [23] also deduced that the strength of porous vegetarian concrete is governed by the water cement ratio and cement content, simultaneously. Permeability is increased with any increment in aggregate gradation and any decrease in cement paste content. The thickness of the concrete blocks and topsoil affect the growth of plants.

However, there were some application issues with the use of recycled aggregates in PVC. Güneyisi at first noted that the properties of porous concrete were significantly affected by using recycled aggregate. The replacement of natural aggregate adversely influenced the mechanical properties of such concretes [24]. The cracking and nonuniformity of mortar caused by segregation may lead to the failure of constructions, especially in submerging engineering environments [25–27]. Hence, the previous study introduced many solutions for reducing cracking and resisting segregation in porous concrete. Zaetang et al. [28] studied incorporation in porous concretes by replacing the natural aggregate at different levels and discovered that the improvements in strength and abrasion resistance were achieved as a result of better bonding between the recycled aggregate and cement paste due to increased surface porosity and roughness of recycled aggregates. Galishnikova et al. [29] reported that a practical way to utilize a high percentage of recycled aggregate in concrete is by incorporating 25–35% of fly ash since some of the drawbacks induced by the use of recycled aggregates in concrete could be minimized. Kazmi studied the effect of different aggregate treatment techniques on the freeze-thaw and sulfate resistance of recycled aggregate concrete. The results showed that the durability performance of concrete can be estimated through the physical properties of aggregates leading to the durable and eco-friendly design of concrete structures [30]. Wu et al. [31] performed a laboratory investigation on the cracking of recycled concrete, providing an acceptable way to reduce the behavior of cracking in fresh concrete with waste fiber addition.

Above all, the strength and vegetarian properties of porous concrete materials depend on the micro and meso scale structure of porous concrete, especially in the transition zone between RAC and cement. Hence, enhancing the strength of mortar around RAC is a

feasible way to increase the mechanical behavior and environmental performance of porous concrete while using RAC. To eliminate the effect of segregation and cracking phenomenon in fresh porous concrete, many works have been conducted in previous research. In this paper, a simple way to modify the properties of PVC is provided by adding hydroxypropyl methylcellulose (HPMC) as an additive. In order to estimate the properties of fresh mortar and concrete while using a different dosage of HPMC, a comprehensive experimental investigation of the viscocit, flowability, strength, durability, and vegetarian properties was performed. The feasibility of using HPMC is determined by the results of the cement and concrete test.

## 2. Experiment

Porous vegetarian concrete (PVC) is a composite material combined with vegetarian filling and the porous concrete skeleton. The porous concrete comprises recycled aggregate bonded together with cement mortar. The filling materials contain planting solid and packing seed. Hence, the influences of a varied HPMC ratio on the performance of PVC is divided into three parts in this paper: cement mortar, porous concrete, and PVC. The schematic diagram of PVC is plotted in Figure 1.

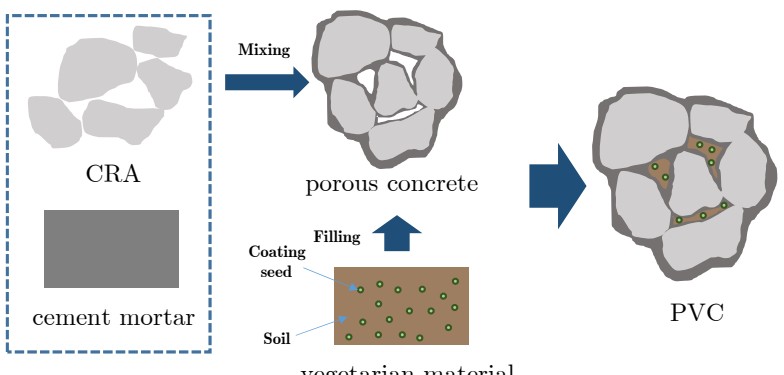

**Figure 1.** Schematic diagram of porous vegetarian concrete (PVC).

### 2.1. Preparation of the PVC Materials

**Cement**: The ordinary Portland cement, P.O 42.5 in Chinese Standard produced by Hailuo Cement Co® (Wuhu, China), was used for the investigation. The specific surface area of the cement was 350 m$^2$/kg.

**Coarse recycled aggregates(CRA)**: The CRA was collected from an urban renewal program in Suining street, Xuzhou, China, produced by Zhenning Industry (Xuzhou, China). The CRA was made by breaking the concrete blocks with a gravity crusher and selected according to the particle size. The porosity of CRA is 24.7% investigated by the Archimedes method. The crushing value index of CRA is 24–28% according to the China National Standard GB/T 14685-2011. The powder content of CRA is approximately 18–24%, which depends on the source of the blocked concrete.

**Admixtures**: The hydroxypropyl methylcellulose (HPMC) was selected as an additive in the cement mortar and porous concrete, which the GOMAZ Chemical Industry supplied. The viscosity of HPMC tested by the Brookfield Viscometer tester was 54,000. The mixing dosage of HPMC is based on the mass of all concrete materials.

A strict procedure was carried out to obtain a consistent performance, described as follows:

1. Add 10 percent by weight of water to the CRA and mix for 3 min;
2. Add cement to the mixture from the previous step and mix for 30 s until aggregates are completely wrapped;
3. Add water and HPMC and mix continuously for 3 min until aggregates are wrapped entirely;

4.   Move the cubic specimens to a greenhouse after 28 days of curing, spread dry soil on the top of the sample, and water slightly. Spread the packing seeds onto the topsoil and water every three days.

*2.2. Test Procedures and Methods*

2.2.1. Performance of the Cement Mortar

The essential performance, such as density and air content, were conducted at first according to the Material Standard JCT601-2010 and GB/T50080-2002, respectively. After that, each fresh sample was poured into the mold and placed on a horizontal flow table as per the mixing design. After tamping 25 times, the mold was vertically lifted and the concrete was allowed to stand on its own without any support. Then the spread of the mortar was measured, in diameter, using a centimeter-scale horizontally ($D_1$) and vertically ($D_2$). Then the value of flowability ($f_m$) was obtained by the formula $(D_1 + D_2)/2$. Finally, the index of viscosity was evaluated by the Brookfield Digital Viscometer tester.

2.2.2. Effect of HPMC on Porous Concrete Properties

The unconfined compressive test was conducted on the cubic specimens ($15 \times 15 \times 15$ cm) to investigate the compressive strength. Table 1 shows the mixture proportions for porous concrete, where four groups of concrete are produced with different dosages of HPMC at a w/c of 0.4.

**Table 1.** Mixing proportions of the porous concretes.

|        | **Cement** | **RCA** | **w/c** | **HPMC** |
|--------|------------|---------|---------|----------|
| PC-0   | 1          | 2.2     | 0.4     | 0        |
| PC-1   | 1          | 2.2     | 0.4     | 1        |
| PC-2   | 1          | 2.2     | 0.4     | 2        |
| PC-3   | 1          | 2.2     | 0.4     | 3        |

The effective porosity test was carried out. The hardened volume $V_a$ could be measured by using the Archimedes method. The effective porosity $P_{eff}$ was determined by

$$P_{eff} = (1 - \frac{V_a}{15 \times 15 \times H}) \times 100\%. \tag{1}$$

where $H$ is the height of hardened specimen. Above that, wetting and drying (w-d) recycling tests (GB/T50082-2009) were carried out for the produced porous concrete samples after 28 days of curing. Each cycle consists of two wetting stages and a drying stage. In the wetting stage, the samples were placed into a tank filled with portable water for 5 h at room temperature of $20 \pm 2$, while in the drying stage, the samples were moved into an oven at $70 \pm 2\,^\circ$C for 42 h. Then the samples were moved to the water tank again after cooling for approximately 5 h. The above procedures were repeated to start another circle. The weight of the samples was frequently measured at the interval of each w-d cycle to determine mass transportation. A compressive strength test was carried out for the samples with different HPMC dosages subjected to w-d cycles of 1, 3, 6, 9, 12, 15 and 20.

2.2.3. Effect of HPMC on Vegetarian Properties of PVC

Vegetarian property is another principle performance for PVC. The porous samples (PC-0, PC-1, PC-2, and PC-3) were filled with 125 packing seeds and solid (PVC-0, PVC-1, PVC-2, and PVC-3, respectively). The plants' height, sprouting number, and root length were recorded at the ages of 3, 7, 15, and 20 days, respectively. In addition, a control group was set to eliminate the disturbance of the soil and seed. The control group was designed with the same number of packing seeds and the same environment of solid and watering at the same time. With the same condition, the sprouting number, height, and root length of the control group were recorded. The relative values in PVC-0, PVC-1, PVC-2, and PVC-3

were measured by the followed functions, and the relative values of plant height($\widetilde{H}_{0,1,2,3}$) are given by

$$\widetilde{H}_{0,1,2,3} = \frac{H_{0,1,2,3}}{H_c} \times 100\% \tag{2}$$

where the $H_{0,1,2,3}$ and $H_c$ are the measured values of PVC-0, 1, 2, 3 and the control group at the same age. The relative values of the sprouting number of PVC-0, PVC-1, PVC-2 and PVC-3($\widetilde{S}_{0,1,2,3}$) are given by

$$\widetilde{S}_{0,1,2,3} = \frac{S_{0,1,2,3}}{S_c} \times 100\% \tag{3}$$

where the $S_{0,1,2,3}$ and $S_c$ are the sprouting number of PVC-0, 1, 2, 3 and the control group at the same age. The relative values of the root length of PVC-0, PVC-1, PVC-2 and PVC-3 ($\widetilde{L}_{0,1,2,3}$) are given by

$$\widetilde{L}_{0,1,2,3} = \frac{L_{0,1,2,3}}{L_c} \times 100\% \tag{4}$$

where the $L_{0,1,2,3}$, and $L_c$ are the measured values of PVC-0, 1, 2, 3 and the control group at the same age.

## 3. Results

### 3.1. Principle Properties of Cement Mortar

The results are listed in Table 2. With the w/c ratio decreasing, the compressive strength of cement mortar increased to the peak value of 26.5 MPa. There was no evidence proving that HPMC contributed to the strength of the mortar directly.

**Table 2.** Mixing proportions of the cement mortars and test results.

| Test Item | Cement | w/c | HPMC (‰₀) | Density ($D_m$, g/cm³) | Air Content (%) | Strength (MPa) |
|-----------|--------|------|-----------|------------------------|-----------------|----------------|
| CE-1  | 1 | 0.35 | 1 | 1.83 | 1.24 | 26.5 |
| CE-2  | 1 | 0.35 | 2 | 1.81 | 1.28 | 26.1 |
| CE-3  | 1 | 0.35 | 3 | 1.81 | 1.44 | 24.8 |
| CE-4  | 1 | 0.4  | 1 | 1.79 | 1.26 | 25.2 |
| CE-5  | 1 | 0.4  | 2 | 1.79 | 1.28 | 25.4 |
| CE-6  | 1 | 0.4  | 3 | 1.77 | 1.35 | 24.8 |
| CE-7  | 1 | 0.45 | 1 | 1.77 | 1.27 | 17.5 |
| CE-8  | 1 | 0.45 | 2 | 1.76 | 1.27 | 18.2 |
| CE-9  | 1 | 0.45 | 3 | 1.74 | 1.32 | 17.7 |
| CE-10 | 1 | 0.5  | 1 | 1.71 | 1.24 | 12.4 |
| CE-11 | 1 | 0.5  | 2 | 1.67 | 1.26 | 13.1 |
| CE-12 | 1 | 0.5  | 3 | 1.69 | 1.32 | 12.6 |

Results of density ($D_m$) and air content are plotted in Figure 2. The average densities of mortar range from 1.67 g/cm³ to 1.83 g/cm³, which had a negative relationship with the w/c ratio significantly. In particular, Figure 2b shows that the dosage of HPMC has a more significant influence on the air content result, which is more noticeable when the w/c is below 0.4. This result was also proved by previous research [32,33].

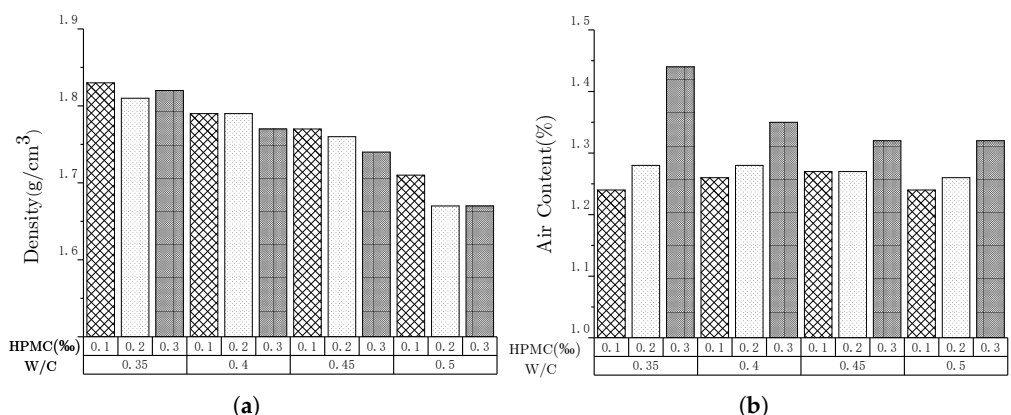

**Figure 2.** Different w/c ratios and HPMC dosages expressed in densities and air contents of fresh mortar. (**a**) Densities of fresh cement mortars; (**b**) air contents of fresh cement mortars.

The influences of HPMC on the flow properties of fresh cement mortars are described in Figure 3. At a constant w/c (0.4), adding HPMC causes a decrease in the mortar flow and an increase in viscosity. It is noted that the curves of the flowability and viscosity versus time can be divided into the initial hydration stage and the accelerated hydration stage. We also notice that as the dosage of HPMC increased, the beginning of hydration was delayed simultaneously, which was quite similar in both the flowability and viscosity results. At the very beginning of the initial hydration stage, the increase of HPMC from 0.0‰ to 0.3‰ of the concrete mass resulted in a significant effect on both the flowability and viscosity of fresh cement mortars. The dosage of 0.3‰ of HPMC significantly affected the water demand in a w/c ratio of 0.4. The silicate reduced the influence of flowability on HPMC in cement hydrating. However, this kind of phenomenon did not exist in the viscosity results, which the Bingham model can explain with sufficient accuracy [34]. The flowability value was dependent on the shear stress of the fluid, which was affected by the viscosity and shear rate together.

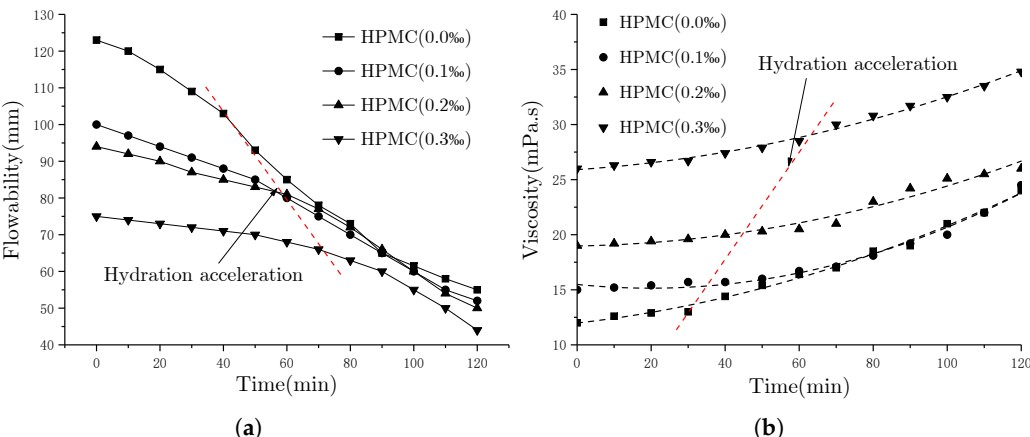

**Figure 3.** Flow properties of the fresh cement mortars at a w/c ratio of 0.4. (**a**) Flowability curves of fresh cement; (**b**) viscosity curves of fresh cement.

### 3.2. Properties of the Hardened Porous Concrete

The effect of the HPMC dosage on the molding form of porous concrete is shown in Figure 4. Compared with group 0.0‰, the deposition phenomenon caused by gravity can be reduced by HPMC added. The HPMC could enhance the viscosity of mortar in the previous results, which means the packing property of the cement grout was also improved in the same w/c and m/a conditions.

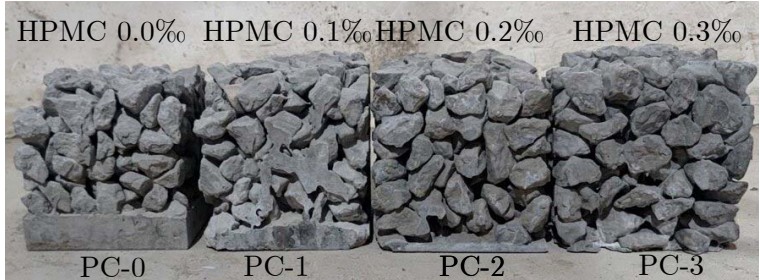

**Figure 4.** Hardened porous concretes with different dosages of HPMC.

The results of the concrete strength are shown in Figure 4. The strength of the filling soil could be negligible compared with the skeleton. The results exhibit that a variable amount of HPMC can improve the strength of PVCs. The curves of all groups were similar to each other, 50% of final strength was reached at the first 3 days and 90% at the age of 7 days, and after that, the strength remained stable. When comparing the compressive strength in the 28 days hydration process, the concrete without any HPMC shows the lowest value in Figure 5a. There was about 35% of strength increasing only by the dosage of HPMC which increased to 2‰. However, changing the dosage of HPMC from 2‰ to 3‰ had no significant effect. The result of the effective porosity (Figure 5b) shows volume expansion in each RAC group at an early stage. In this case, it is difficult to conclude a clear relationship between effective porosity and strength, as in previous observations.

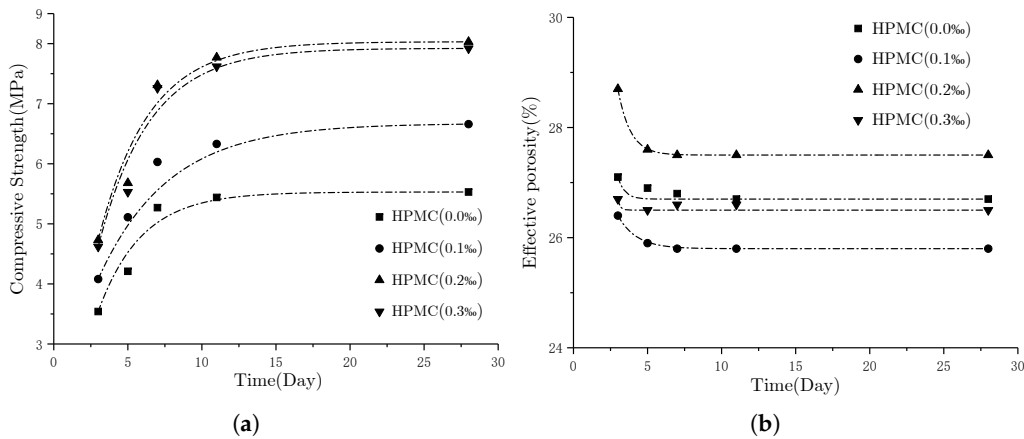

(a)                    (b)

**Figure 5.** Relationships of strengths and porosities versus time for the porous concrete with different dosages of HPMC. (**a**) Strength–time curves for PC-0, 1, 2, and 3; (**b**) porosity–time curves for PC-0, 1, 2, and 3.

Porous concrete subjected to w-d regimes has a prominent influence on the performance. After 28 curing days, the produced samples (PC0-4 in Table 1) were subjected to 1, 3, 6, 9, 12, 15 and 20 w-d cycles. The mass ($M_t$) transportation and compressive strength evolutions with w-d cycles are shown in Figure 6a,b, respectively. Figure 6a shows that with the number of w-d cycles increasing during the 44 days in the test period, $M_t$ increases stably after 12 cycles (PC1-3), like the compressive strength. As the w-d cycle number increases, the general tendency for the strength decreases (see Figure 6b). At the end of 20 cycles, the strength values are approximately 25–45% lower than the corresponding initial ones without the w-d cycle. Note that the influence of strength and $M_t$ by the w-d cycles are more significant in the sample PC-0, which is in good agreement with the viscosity of the cement mortar. Due to adding HPMC, bonding forces are enhanced between recycle courses aggregates [34,35]. Overall, mass transportation is related to the compressive strength of the porous concrete. The material strength is susceptible to the w-d cycle, which could be reduced by adding HPMC.

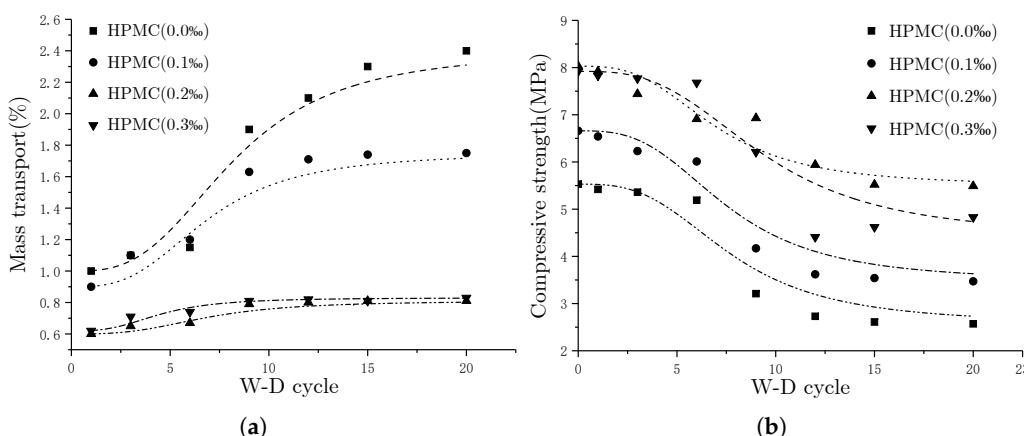

**Figure 6.** The variations of the mass transport and compressive strength of the porous concrete versus w-d cycles. (**a**) Mass transport versus w-d cycles; (**b**) Compressive strength versus w-d cycles.

### 3.3. Vegetarian Properties of the Hardened Porous Concrete

Figure 7 shows the photographs of vegetation in the four types of PVC samples at the age of 14 days and water planting after 28 days, where packing seeds and planting solid were filled in PC-0, PC-1, PC-2, and PC-3, labeled by PVC-0, PVC-1, PVC-2, and PVC-3, respectively. The emerged ratio and plant height were significantly lower in PVC-0 when compared with the vegetation in PVC-1, PVC-2, and PVC-3.

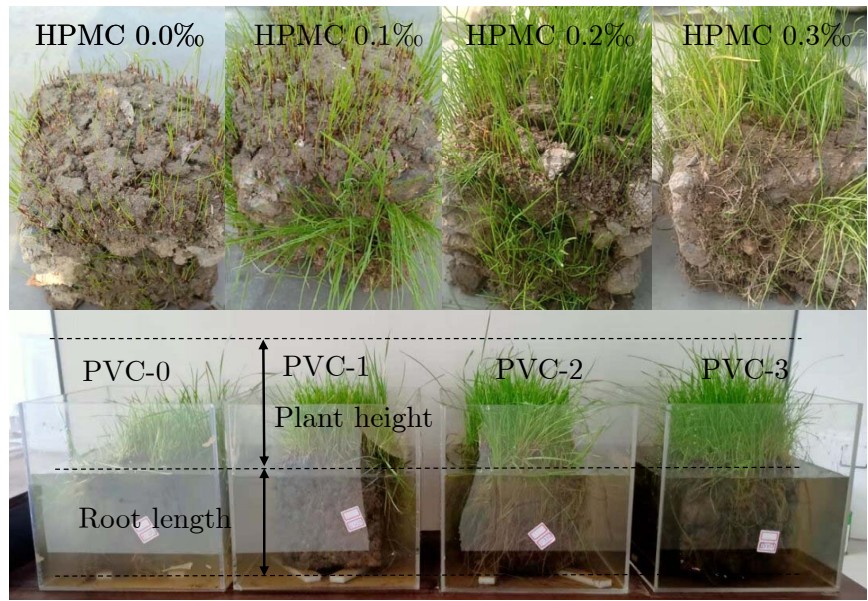

**Figure 7.** Photographs of vegetation in the four types of PVC samples at 14 days and water planting after 28 days.

Figure 8 clearly shows that the vegetarianism in PVC-0 developed slowly both in the sprout ratio and sprout time. It is noticed that with the packing material dissolving, mass of organic constituent, soluble salt and spores of probiotics for plant immersed in soil, the dissolving process benefited plant growth in the control group. However, this kind of situation may not contribute to group PVC-0 and PVC-1. Mostly because the narrow channel and unconnected void in PVC-0 and PVC-1 cannot support sufficient space for water retained, the result is that the packing materials around the seeds is not soluble at room temperature and forms a gel-like solution. From these appearances, it can be deduced that the packing material around the seed cannot be dissolved in a filling solid because of

the narrow space in PC-0. This phenomenon was also be recommended in the previous research [36–38].

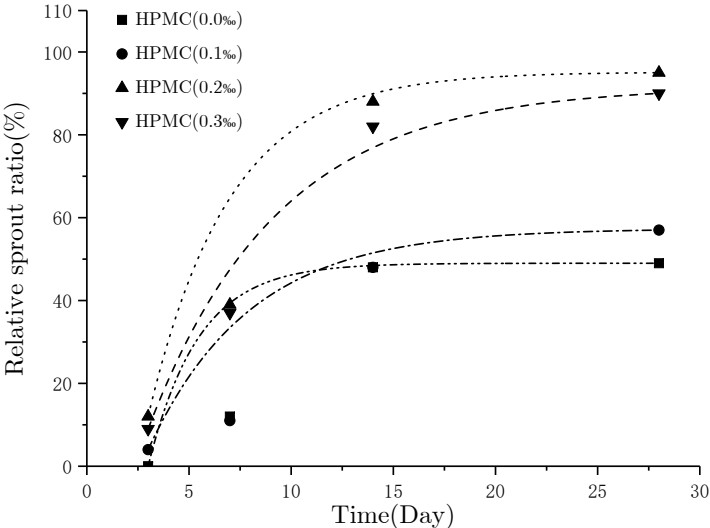

**Figure 8.** Curves of relative emerged versus time.

The results of the relative height and root length are fairly similar to each other in Figure 9a,b. Figure 9 describes that PVC-2 and PVC-3 are most suitable for planting compared with PVC-0 and PVC-1. We noticed that the maximum root lengths of PVC-2 and PVC-3 at 28 days are 20% greater than those of the control group. It is probably because the PVC-2 and PVC-3 are affected by light compared with the control group underground and void structure of the porous concrete. Overall, the influence of vegetarian properties seems more sensitive to the HPMC change, and it can be concluded that with HPMC added, the development condition of the plant is improved. However, the degree of influence of the HPMC on the void structure is still not well understood.

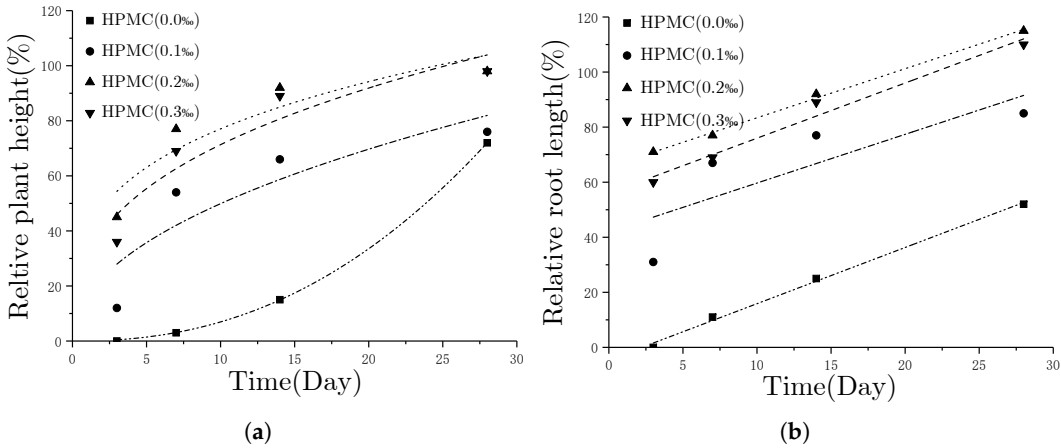

**Figure 9.** Curves of vegetarian development in PVC-0, PVC-1, PVC-2, and PVC-3. (**a**) Relative height of the plant versus time; (**b**) relative root length of the plant versus time.

## 4. Conclusions

This study focused on the feasibility of using HPMC in porous vegetarian concrete which contains RACs. The effects of different dosage of HPMC on the viscocity and flowability of fresh mortar were first investigated. In addition, the strength, durability, effective porosity were explored, and the vegetarian properties such as relative sprout ratio, root length, plant height were verified by the control group method. The effect regarding

the methods used, the feasibility of using HPMC in PVC can be characterized either by the mortar properties or mechanical and vegetarian performance of porous concrete. The main conclusions drawn are summarized as follows:

- The different HPMCs influence the rheology in a remarkable way. However, the effects of the density and air content of the cement mortar are influenced more by changes in the w/c ratio than the admixtures. At a constant w/c of 0.4, HPMC decreases the mortar flow and increases viscosity, while such an accelerated hydration time is also delayed. The influence of the porous concrete performance is similarly affected by the varied HPMC. The bonding forces between the CRA particles and aggregates packing-layer are elevated by the viscosity improvement. The phenomenon of segregation caused by the CRA powder in porous concrete could also be relieved by adding HPMC. Furthermore, these effects caused by the addition of HPMC can improve the durability of porous concrete. The strength is susceptible to wetting–drying cycles, which HPMC could reduce.

- The influence of the porous concrete performance is similarly affected by varied HPMC. The bonding forces between CRA particles and aggregates packing-layer were elevated by the viscosity improvement. The strength was about 35% increased by HPMC increasing to 0.2‰ after 12 days curing. The hardened porous concrete showed that the phenomenon of segregation caused by the CRA powder in porous concrete can also be relieved by adding HPMC. A relatively prominent fluctuation in mass transport and compressive strength with the w-d cycle number occurs before 12 cycles, after which they become stable, except PVC-0. As the ratio of HPMC reaches 0.2‰, the strength and durability becomes susceptible to wetting–drying cycles, which HPMC could reduce. It can be deduced that the durability of porous concrete in the w-d condition is improved by HPMC. The porosity of PVC-0, 1, 2, and 3 was stable at 26.9%, 25.9%, 27.6%, and 26.5% after 7 days curing, respectively. Although the porosity was increased in some cases, the strength was not affected in the compressive test, which is beneficial for HPMC used in PVC. This kind of mechanical property is also presented in other discrete particle materials [39,40].

- By the void structure changing, the vegetarian properties is improved by adding HPMC. The sprout time, root length, and plant height were accelerated by both void structure improvements. The vegetation growth in PVC-0 was almost 20 days slower when compared with PVC-2, and this phenomenon was more remarkable in the results of the sprout ratio test. It can be deduced that with adding HPMC, the void structure becomes more suitable for plant growth.

In general, the principle factor of strength was more influenced by forces between particles than the porosity [41]. On the other hand, sufficient space in porous concrete is more suitable for plant growth due to a large number of pore channels that exist. The studied HPMC can be used while the powder content is unstable due to the source of CRA changing; the benefits of HPMC were significantly proved by the results of the laboratory tests and engineering application. HPMC enables the production of mortars and PVCs at a sufficient strength for engineering with the outstanding void structure, which plays an essential role in controlling the flow and mechanical properties of mortar and porous concrete, respectively. We will try to find an accurate way to describe the strength of this kind of particle-bonding material, which could be used in the optimization design of the PVC mixture, which needs further study.

**Author Contributions:** Data curation, W.Y., X.C. and H.L.; Formal analysis, H.L.; Investigation, X.C. and M.C.; Methodology, W.Y.; Project administration, W.Y.; Resources, K.Z. and S.Z.; Software, K.Z.; Supervision, X.C. and S.Z.; Writing—original draft, K.Z.; Writing—review & editing, W.Y. All authors have read and agreed to the published version of the manuscript.

**Funding:** This research was funded by National Natural Science Foundation of China grant number 51904110 and Natural Science Foundation of the Higher Education Institutions of Jiangsu Province of China grant number 20KJB560006. We would also like to thank the reviewers for their valuable comments that led to an improved manuscript.

**Institutional Review Board Statement:** Not applicable.

**Informed Consent Statement:** Not applicable.

**Data Availability Statement:** Not applicable.

**Conflicts of Interest:** The authors declare no conflict of interest.

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
