# Peer review of "The Feasibility of Hydroxypropyl Methylcellulose as an Admixture for Porous Vegetarian Concrete Using Coarse Recycled Aggregates"

_buildings, doi:10.3390/buildings12050676_

Round 1

Reviewer 1 Report

The authors investigate the potential of hydroxypropyl methylcellulose as an admixture for porous vegetarian concrete made with coarse aggregate. The topic is interesting, and it fits the scope of the journal. The state-of-art is well presented. The research methodology is also well described and appropriate. English use and formatting are good enough. The conclusions are well justified with the results.

The naming of "porous vegetarian concrete" is my major concern in this work. That doesn't sound right to me. Vegetation growing concrete might be the most evocative name.

Minor comment line 116 “…ranging from 1.83g/cm3 to 1.67 g/cm3”. It shall read as follows ““…ranging from 1.67 g/cm3 to 1.83 g/cm3”

Author Response

Reviewer #1:
The authors investigate the potential of hydroxypropyl  methylcellulose as an admixture for porous vegetarian concrete made with coarse aggregate. The topic is interesting, and it fits the scope of the journal. The state-of-art is well presented. The research methodology is also well described and appropriate. English use and formatting are good enough. The conclusions are well justified with the results.
[Answer] Thank you for your positive and pertinent evaluation comments on our paper.
1. The naming of "porous vegetarian concrete" is my major concern in this work. That doesn't sound right to me. Vegetation growing concrete might be the most evocative name.
[Answer] Sorry for the misunderstanding. Vegetarian growing concrete and planting concrete are more suitable for this kind of materials, and vegetarian growing concrete contains serial materials with cement and plants conbined. Some of them are continuum materials(using fabric and soil with few cement bonding materials as planting material), some of them are not, which including prous concrete with vegetarian growing. In this opinion, to highlight he  properties of porous and maintain the consistency of naming(from porous concrete to porous vegetarian concrete). The porous vegetarian concrete was selected in this paper.
2. Minor comment line 116 “…ranging from 1.83 g/cm3 to 1.67 g/cm3”. It shall read as follows ““…ranging from 1.67 g/cm3 to 1.83 g/cm3
[Answer] Sorry for the discribing error, we have checked all the text and corrected these errors.

Reviewer 2 Report

In the article "The feasibility of hydroxypropyl methylcellulose as an admixture for porous vegetarian concrete using coarse recycled aggregates " the effect of the HPMC on the fluidity of the mortar was first investigated by viscosity test. 

The study is interesting and well-written.

There is need to incorporate quantitative results in abstract.

The introduction needs revision to further highlight the importance and novelty of this study.

Table 1. what was the reason for w/c = 0.4, please provide reasons and why other w/c were not considered as well.

Equation 2 should be divided into parts for better clarity.

Line 114. Please provide proper reasons for the observations and also support with previous studies.

Line 126 needs correction.

Why section 4 is needed in this article? is it necessary, it does not show any relation with this research article.

Conclusions must be more concise and to the point.

Author Response

Reviewer #2:
In the article "The feasibility of hydroxypropyl methylcellulose as an admixture for porous vegetarian concrete using coarse recycled aggregates " the effect of the HPMC on the fluidity of the mortar was first investigated by viscosity test. The study is interesting and well-written.
[Answer] Thanks a lot for your comments.
1. There is need to incorporate quantitative results in abstract.
[Answer] Thanks a lot for your comments. The results has been completed.
2. The introduction needs revision to further highlight the importance and novelty of this study.
[Answer] Thank you for your comments and we feel very sorry for the inconvenience brought to the reviewer. The introduction has been improved. We will increase our writing skill as soon as possible.

3. Table 1. what was the reason for w/c = 0.4, please provide reasons and why other w/c were not considered as well.
[Answer] Thank you for your rigorous consideration. Indeed, the w/c was a quite
important parameter in cementitious materials, the w/c effects on the strength of porous concrete were quite sufficient in previous researches (https://doi.org/10.1016/j.conbuildmat.2006.12.007, https://doi.org/10.1016/j.conbuildmat.2011.12.024). Due to the HPMC was not involved in the cement hydration (https://doi.org/10.1016/j.cemconres.2005.08.003), only delayed.
By this reason, we only discussed the effects on the density and air content of w/c changing. But thanks a lot to your comments, the optium w/c ratio will be discussed in our next paper.
4. Equation 2 should be divided into parts for better clarity.
[Answer] Special thanks for your good suggestion. We have made correction according to your comments in our revised version.
5. Line 114. Please provide proper reasons for the observations and also support with previous studies.
[Answer] Thanks a lot for your comments. Xiaowei Gu (Materials 2021, 14(21), 6451; https://doi.org/10.3390/ma14216451) confirmed as the entrained air content of concrete increases, the compressive and flexural strength decreases, and the chloride ion electric flux increases after adding HPMC by MIP test. What was more, Yulin Zhang (CBM, 2016(102); https://doi.org/10.1016/j.conbuildmat.2015.10.116) also proved that HPMC
can introduce additional gas content into the foam concrete, therefore slightly reduce the initial wet density of foam concrete. The above researches had already been added in the reference.
6. Line 126 needs correction.
[Answer] Thanks for your comments. We have made correction according to your comments in our revised version.
7. Why section 4 is needed in this article? is it necessary, it does not show any relation with this research article.
[Answer] Thank you for your positive comments on our paper. Section 4 was removed in this paper.
8. Conclusions must be more concise and to the point.
[Answer] We gratefully appreciate for your valuable comment. We have improved our conclusions.

Reviewer 3 Report

The manuscript deals with the feasibility of hydroxypropyl methylcellulose as an admixture for porous vegetarian concrete using coarse recycled aggregates .

 I recommend the article for publishing after taking into account the following remarks:

  1. The article states that “the effect of the HPMC on the strength and durability of the hardened PVC” will be analysed.

Chapter 3.2 presents the properties of the hardened porous concrete, example the compressive strength. However, key test results related to the durability of the material, such as frost resistance and water absorption, are missing.

Please add the research results, if the authors have done such research.

  1. The introduction is very brief.
  2. Line 150 space is missing “Figs. 6aand”

Author Response

Reviewer #3:

The manuscript deals with the feasibility of hydroxypropyl methylcellulose as an admixture for porous vegetarian concrete using coarse recycled aggregates .  I recommend the article for publishing after taking into account the following remarks:

[Answer] We thank the reviewer for reading our paper carefully and giving the above positive comments.

  1. The article states that “the effect of the HPMC on the strength and durability of the hardened PVC” will be analysed. Chapter 3.2 presents the properties of the hardened porous concrete, example the compressive strength. However, key test results related to the durability of the material, such as frost resistance and water absorption, are missing. Please add the research results, if the authors have done such research.

[Answer]Yes, because this kind of materials was designed in the riparian buffer area, such as bank slope in water condition, so the main focus of this manuscript is on the materials in W-D condition. But thanks a lot to your comments, the frost resistance and water absorption will be the major topic in our next paper.

  1. The introduction is very brief.

[Answer] Thanks a lot for your comments. The introduction part has be improved.

  1. Line 150 space is missing “Figs. 6 aand”

[Answer] Sorry for the spelling error, we have checked all the text and corrected these errors.

Round 2

Reviewer 1 Report

The authors addressed all of the comments, and the article can be published as is.